# Chromogenic Chemodosimeter Based on Capped Silica Particles to Detect Spermine and Spermidine

**DOI:** 10.3390/nano11030818

**Published:** 2021-03-23

**Authors:** Mariana Barros, Alejandro López-Carrasco, Pedro Amorós, Salvador Gil, Pablo Gaviña, Margarita Parra, Jamal El Haskouri, Maria Carmen Terencio, Ana M. Costero

**Affiliations:** 1Instituto Interuniversitario de Investigación de Reconocimiento Molecular y Desarrollo Tecnológico (IDM), Universitad Politècnica de València, Universitat de València, Doctor Moliner 50, Burjassot, 46100 Valencia, Spain; Mariana.Barros@uv.es (M.B.); alejandro.lopez-carrasco@uv.es (A.L.-C.); salvador.gil@uv.es (S.G.); Pablo.Gavina@uv.es (P.G.); Margarita.Parra@uv.es (M.P.); Carmen.Terencio@uv.es (M.C.T.); 2Instituto de Ciencia de Materiales (ICMUV), Universitat de València, P.O. Box 2085, 46071 Valencia, Spain; Jamal.Haskouri@uv.es; 3CIBER de Bioingeniería, Biomateriales y Nanomedicina (CIBER-BBN), 28029 Madrid, Spain; 4Departamento de Farmacología, Universitat de València, Vicente Andrés Estellés S/n, Burjassot, 46100 Valencia, Spain

**Keywords:** detection, spermine, spermidine, silica particles

## Abstract

A new hybrid organic–inorganic material for sensing spermine (Spm) and spermidine (Spd) has been prepared and characterized. The material is based on MCM-41 particles functionalized with an N-hydroxysuccinimide derivative and loaded with Rhodamine 6G. The cargo is kept inside the porous material due to the formation of a double layer of organic matter. The inner layer is covalently bound to the silica particles, while the external layer is formed through hydrogen and hydrophobic interactions. The limits of detection determined by fluorimetric titration are 27 µM and 45 µM for Spm and Spd, respectively. The sensor remains silent in the presence of other biologically important amines and is able to detect Spm and Spd in both aqueous solution and cells.

## 1. Introduction

Under normal physiological conditions, the concentration of polyamines in the body is kept constant through a complex mechanism that involves biosynthetic, catabolic, and transport processes [1,2,3]. An increase in intracellular polyamine concentration can be correlated with uncontrolled cell proliferation and tumorigenic transformation [4,5,6,7]. Thus, high levels of polyamines have been related to skin, prostate, colon, or breast cancers [8,9,10,11,12,13]. Among the polyamines, spermidine (Spd) and spermine (Spm) (Figure 1) have been proven to be interesting biomarkers, in tissues and biological fluids, for detecting diverse pathological situations. Hence, it has been demonstrated that low- and high-grade prostate cancer tissues can be distinguished by the spermine concentrations among other metabolites [14]. Moreover, the concentrations of these amines in biological fluids have been used to detect other pathological conditions. For example, it has been demonstrated that the concentration of Spd + Spm in psoriasis patients is around 5.5-fold higher than in healthy individuals [15]. By contrast, it has been also established that a low concentration of these amines can be associated with aging related illnesses. In fact, the total Spd + Spm concentration in the blood is lower in 60–80-year-old individuals than in 31–56-year-old individuals [16]. For this reason, these polyamines have been considered promising biomarkers for the early detection of aging related illnesses, such as Parkinson’s disease [17]. Therefore, spermidine seems to play an important role in memory and longevity [18]. For these reasons, the detection of these polyamines is of great interest, and the design and synthesis of probes able to detect their presence in biological fluids, such as urine, and in tissues are a constantly developing field.

In addition to the traditional spectroscopic methods for detecting Spm and Spd such as MALDI–TOF MS [19], GC-MS [20], or Electrospray Ionization and Time-Of-Flight Mass [21], the use of chromo- or fluorogenic probes is an active research field due to the advantages of these types of sensors, such as no need of expensive instrumental, real-time detection or easy preparation and use. Some of these probes are based on organic molecules that experiment changes in fluorescence color after reacting with the polyamines [22,23,24], whereas other protocols are based on the use of metallic or organic nanoparticles [25,26,27] (for more details see Appendix A in the Appendix A). However, even though mesoporous silica nanoparticles can also be used in the preparation of chromogenic or fluorogenic sensors [28], there is only one example in which this type of nanoparticles is used for detecting Spm and Spd [29] and, even in this case, the silica particles are decorated with Au that is responsible for the transduction mechanism. What is more, at the best of our knowledge, there are no examples of Spm or Spd detection using silica nanoparticles capped with molecular gates. Silica mesoporous supports are often used as containers because of their interesting properties: well-known chemistry, high inertness, easy functionalization, large surface, and specific volume, amongst others [30,31]. More specifically, the use of organic–inorganic hybrid materials has been demonstrated to offer many applications. Amid these materials, silica nanoparticles capped with molecular gates have shown to be very useful not only in the delivery process but also in chemical detection. Apart from the inorganic scaffold these gated nanodevices contain, a switchable “gate-like” ensemble capable of being “opened” or “closed” when certain external stimuli are applied. In this way, the silica scaffold works as a container in which an appropriate dye or fluorescent molecule is loaded. The molecular gate keeps the material inside this container until the adequate analyte (that works as a specific stimulus) “opens” the gate allowing the indicator (dye or fluorophore) to be released and observed in solution [32,33,34].

Based on our experience in chemo-sensing [35,36,37] and in the use of hybrid organic–inorganic nanomaterials functionalized with molecular gates for drug releasing and sensing [38,39,40], we herein report a new sensor able to detect both Spm and Spd in solutions and cells using an unexplored protocol based on silica particles capped with molecular gates. Moreover, the limit of detection determined for these amines are low enough to detect pathologic situations, such as ovarian tumors, whereby the concentration in urine is around 40 µM [41,42].

The sensing protocol is described in Scheme 1. It is based on the use of MCM-41 particles loaded with a dye (Rhodamine 6G) and capped with an N-hydroxysuccinimide derivative 1. In this hybrid material (S1), particles present an internal shell in which derivative 1 is covalently bound to the silanol groups, and an external sphere of additional N-hydroxysuccinimide derivatives is kept together through hydrogen bond and hydrophobic interactions [43]. This covering keeps the dye inside the pores. In analyte presence, the amine groups remove the external coverage due to its stronger acid-base interactions with the N-hydroxysuccinimide derivative placed in the external shell. The removal of this external coverage opens the pores allowing the dye to release.

## 2. Materials and Methods

### 2.1. Chemicals and Materials

The reagents employed in the synthesis were acquired from Sigma Aldrich (Darmstadt, Germany) and used without further purification. ^1^H NMR and ^13^C NMR spectra were registered with Bruker Avance 300 MHz, 400 MHz, or 500 MHz spectrophotometers (Billerica, MA, USA), all of them referenced to solvent peak, MeCN(d_3_). UV-Vis spectra were registered with a Shimadzu UV-2600 spectrophotometer (Nakagyo-ku, Kyoto, Japan), using a 1 cm of path length cuvette. Fluorescence measures were carried out with a Cary Eclipse Spectrofluorometer (Santa Clara, CA, USA), using a 1 cm of path length cuvette. The solids were characterized by X-ray powder diffraction (XRD) at low angles (Seifert 3000TT θ−θ) using Cu Κα radiation. Patterns (Maitenbeth, Germany) were collected in steps of 0.03° (2θ) over the angular range 0.73–10° (2θ) for 10 s per step. An electron microscopy study (HRTEM) was carried out with a JEOL JEM-1010 (Akishima, Tokio, Japan) instrument operating at 100 kV. The N_2_ adsorption–desorption was recorded with a Micromeritics ASAP2020 (Norcross, GA, USA) automated sorption analyzer. Samples were degassed at 120 °C in vacuum overnight. Specific surface areas were calculated from the absorption data in the low-pressure range using the Brunauer–Emmett–Teller (BET) model. Pore size was determined following the BJH method. Dynamic light scattering (DLS) measurements were performed using Malvern Nanosizer ZS equipment (Malvern, UK). Samples were suspended in water and treated in an ultrasound bath during some minutes before measuring. Wallac 1420 VICTOR2™, PerkinElmer (Boston, MA, USA).

### 2.2. Synthesis of 1-Hydroxy-3-((3-(trimethoxysilyl)propyl)thio)pyrrolidine-2,5-dione (1)

N-hydroxymaleimide (113 mg, 1 mmol) was dissolved in acetonitrile (12 mL) containing imidazole (6 mg). To the stirred solution, 3-(trimethoxysilyl)propane-1-thiol (190 µL, 1 mmol) was added. After stirring overnight at room temperature, the solvent was evaporated, and the residue (292 mg, 94%) was characterized and used in the following step.

^1^H-NMR CD_3_CN (500 MHz) 3.77 (1H, dd, *J* = 9.0, 3.6 Hz, H-3); 3.52 (9H, s, OMe); 3.11 (1H, dd, *J* = 18, 9 Hz, H-4c); 2.80 (1H, ddd, *J* = 12.7, 7.8, 7.0; H-1′); 2.70 (1H, dt, *J* = 12.7, 7.5 Hz; H-1′); 2.46 (1H, dd, *J* = 18, 3.6 Hz, H-4t); 1.67 (2H, m, H-2′); 0.73 (2H, t *J* = 8.4 Hz, H-3′). ^13^C-NMR CD_3_CN (125MHz) 172.6 and 170.7 (2 N-C=O), 50.8 (O-Me), 37.9 (C-3), 34.7, 34.5, 23.5 (C-2′), 9.0 (C-3′).

### 2.3. Synthesis of Mesoporous Silica Particles (MCM-41)

CTABr (2 g, 5.48 mmol) was dissolved in deionized water (960 mL) and 2 M aqueous NaOH (7 mL) were added until pH 2 was attained in the solution, which was heated. When 80 °C reaching, TEOS (10 mL, 51.4 mmol) was added dropwise under strong stirring. The mixture was left at 80 °C and stirred for an additional 2 h period. A white precipitate was obtained, which was isolated by centrifugation and washed with deionized water and ethanol until a neutral pH was reached. Afterwards, it was dried in a stove at 60 °C until a constant weight was attained (2.68 g). The surfactant was eliminated by calcination at 550 °C under an oxidant atmosphere for 5 h yielding the MCM-41 (1.37 g).

Preparation of S1: MCM-41 (200 mg) and Rhodamine 6G (200 mg) were suspended in dry CH_3_CN (35 mL), and the mixture was stirred under an Ar atmosphere for 24 h, at room temperature.

Procedure 1: Compound **1** was dissolved in a mixture of CH_3_CN and DMSO (5:1) and then added to the reaction, and the mixture was stirred for 24 h. Finally, the material was washed with acetonitrile, water, and ethanol, and dried on the stove at 50 °C.

Procedure 2: 3-(trimethoxysilyl)propane-1-thiol (260 µL, 275 mg, 1.4 mmol) was added, and the mixture was stirred at room temperature for 5.5 h, before the addition of N-hydroxymaleimide (148 mg, 1.4 mmol) and imidazole (10 mg), and then the mixture was stirred overnight. Finally, the material was washed with acetonitrile, water, and ethanol and dried on the stove at 60 °C.

### 2.4. Sensing Experiments

The response of S1 particles was tested in the absence and presence of amines by monitoring the release of the trapped Rhodamine 6G from S1 (λ_ex_ = 525 nm and λ_em_ = 550 nm). In a typical experiment, 1 mg of solid S1 was suspended in 300 µL of an aqueous solution of amines with a concentration of 10^−2^ M (spermine, spermidine, and putrescine), then this suspension was diluted with 2700 µL of PBS buffer (pH = 7.4) to obtain a 10^−3^ M concentration of amines.

### 2.5. Disaggregation Experiments

To determine the amount of loaded Rhodamine 6G in S1 material, 1 mg of S1 was dissolved in 3 mL of an aqueous solution of NaOH (0.5 M) with 10% MeOH, and after 24 h, the absorbance and fluorescence of the solutions were measured, and the concentration was evaluated using the calibration curves.

## 3. Results and Discussion

### 3.1. Synthesis and Characterization of Probe S1

For the synthesis of probe S1, mesoporous silica particles of the MCM-41 type were synthesized using cetyltrimethylammonium bromide as a structure-directing agent and tetraethylorthosilicate as a silica source [44,45]. The pores of the inorganic scaffold were loaded with Rhodamine 6G, and then the external surface was functionalized following two different approaches. In a first approach, compound **1** was synthesized, as shown in Scheme 2, from N-hydroxymaleimide and 3-(trimethoxysilyl)propane-1-thiol in the presence of imidazole and then bound to the surface of the MCM-41 particles loaded with Rhodamine 6G. The second approach followed a two steps synthesis. First, MCM-41 loaded with Rhodamine 6G was reacted with 3-(trimethoxysilyl)propane-1-thiol, and then the solid was submitted to reaction with N-hydroxymaleimide in the presence of imidazole to produce the designed material. Both procedures led to very similar materials. The main difference between both procedures is the increment in cargo observed in the second procedure (1.6% Rhodamine 6G in the first case versus 2.7% in the second one). This can be related to the presence of the thiol groups that favor the loading of the lipophilic dye through Van der Waals interactions as had been reported in the literature [46].

To understand why the cargo is kept entrapped into the pores, as-made MCM-41 particles were treated with compound **1**, and the obtained material was studied by solid ^13^C NMR (see Appendix A in Appendix A). The material showed the expected peaks from the organic moiety, and elemental analysis indicated that around 5% of organic material had been bound to the silica matrix. When similar experiments were carried out with particles charged with Rhodamine 6G, it was observed that around 19% of the organic material corresponded to the molecular gate. The difference between both values suggests that a large amount of organic material is kept surrounding the particles when the dye is present. This is probably due to them interacting by hydrogen bonds or hydrophobic interactions.

### 3.2. Characterization of the Prepared Materials

The as-made MCM-41 silica and solid S1 were characterized using the usual techniques for these materials: powder X-ray diffraction (PXRD), transmission electron microscopy (TEM), N_2_ adsorption–desorption isotherms, and elemental analysis including thermal analysis (TGA, see Appendix A in Appendix A).

The XRD patterns for the two materials show unambiguously the presence of a hexagonally ordered array both before (pure silica) and after the dye charge and the incorporation of the external organic arms of the nanogate ensemble (see Figure 2 for comparison between MCM-41 and S1). In both spectra, an intense and narrow diffraction peak is detected in the 2–3 2θ(°) angular range together with three additional signals of lower intensity (4–7 2θ(°) angular domain) though extremely well resolved. These XRD peaks can be indexed to the (100), (200), (110), and (210) planes, assuming a highly ordered 2D hexagonal unit cell (P6mm), typical of the well-ordered MCM-41 silica [47]. The single difference between the PXRD patterns of the starting pure silica and the final S1 material lies in the relative intensity of the peaks. The dye cargo inside the mesopores and also a certain incorporation of the 3-(trimethoxysilyl)propane-1-thiol groups (during the first step of the nanogate construction) at the mesopore entrances favor the filling of the mesopores with organic species with the ensuing loss of contrast between the silica walls and the pores (that are initially completely voids) [48]. Regardless of this expected tendency, we can remark that the relative intensities of the four observed peaks in each spectrum are similar before and after functionalization, and this supports the complete preservation of a highly ordered mesopore array.

TEM images (Figure 3) confirm the existence of a hexagonal symmetry of the mesoporous array. TEM images show well-ordered white spots associated with the mesoporous that, depending on the relative orientation of the sample respect to the electron beam, result in stripes or ordered spots with hexagonal symmetry associated with compact packing of the cylindrical mesopores. In some cases, this hexagonally ordered organization on the mesopore scale is transferred to the particle shape (see upper inset in Figure 3a and inset in Figure 3b).

The sample morphology can be described as being formed by spherical and pseudospherical mesoporous particles with individual sizes lower than 200 nm: with average sizes of 135 ± 45 and 138 ± 53 nm for MCM-41 and S1 samples, respectively. The starting pure silica shows a high degree of dispersion (see lower inset in Figure 3a). A large proportion of isolated particles (or forming relatively small aggregates) can be observed in TEM images. After the dye loading and the subsequent gate construction, the aggregation degree increases, probably due to the interaction between the organic N-hydroxysuccinimide arms. However, this phenomenon is not a problem when thinking about a sensor, since that the analytes (polyamines in this case) can easily diffuse along the large textural voids among particles. The DLS curves (see Appendix A in Appendix A) show that, after gentle ultrasound treatment, both samples have very similar particle size distributions with values that agree with those estimated from TEM images. In fact, peak mean values of 154.8 and 161.0 nm have been estimated for MCM-41 and S1 samples, respectively; and maximum peak values of 132.4 and 141.8 nm for MCM-41 and S1, respectively.

The porosity of the solids is further illustrated in a quantitative way through N_2_ adsorption–desorption isotherms (Figure 4 and Table 1). The starting silica shows a typical type IV isotherm according to the IUPAC classification, with no hysteresis loop. This is characteristic of well-ordered MCM-41 derivatives with regular mesopores and without pore necking. The filling of mesopores occurs at intermediate relative pressure values in the 0.2–0.4 range as an abrupt increase in the adsorbed volume. The second adsorption step, at P/P_0_ values > 0.9, is due to a residual textural porosity, in the macropore range, which was associated with the voids among primary particles [49]. The pore volume due to this last porosity is clearly lower than that associated with the intra-particle mesoporosity. After functionalization, the resulting S1 solid presents a decrease in surface area as in pore volume when comparing to the starting silica. The BET area and the intra-particle mesopore volume evolve from 1145 to 777 m^2^/g, and from 1.08 to 0.30 cm^3^/g, respectively. This indicates that ca. 66% of the intra-particle mesopores are inaccessible for N_2_ adsorption, and consequently filled with dye molecules. In fact, a small and relatively smooth adsorption step is observed in the isotherm in the low-pressure domain when compared to the parent silica. This step is associated with a low pore volume (0.30 cm^3^/g) and a BJH mesopore of 2.12 nm (markedly lower than that measured for the starting material, 2.96 nm). This decrease suggests that some mesopore entrances could present a low density of functional organic arms or partial construction of the gate assembly (probably due to the sequential formation of the nanogate). In any case, the proportion of non-effective mesopores (ca. 33%) for the signaling process is relatively low and does not suppose any problem. The textural porosity (due to the interparticle voids) is qualitatively preserved, but in the case where the S1 solid seems to be enhanced, probably due to a certain interparticle aggregation (according to the TEM images) associated with the interaction between the organic arms between different particles.

The amount of well-formed nanogates is around 1 mmol/g. This amount is within the range of functionalization values previously described by our group in a great variety of nanogates (0.5–3.5 mmol/g) [50]. However, the amount of Rhodamine 6G is significantly higher (2.7 wt%) than that previously charged in previous works (0.57–0.95 wt% range) [51]. These differences are probably due to both the size of the organic arms and the gate construction method (directly or sequentially). In the present work, we use a sequential method, in which the first stage is carried out with a reagent, 3-(trimethoxysilyl)propane-1-thiol, of relatively small size, which will avoid reactivity problems associated with steric hindrances. This favors rapid functionalization through condensation reactions with the silanol groups on the surface of the silica, which probably avoids the loss of colorant during the subsequent nanogate formation steps. In fact, a certain amount (2.2%) of “free” propane-1-thiol groups is observed in addition to those implied in the complete organic arms by incorporation in the second step of the N-hydroxymaleimide. This amount could be associated to 3-(trimethoxysilyl)propane-1-thiol groups incorporated inside the mesopores (instead of the outer surface and/or mesopore inlets), which would explain the difficulty of these to react in the second stage with the N-hydroxymaleimide. On the contrary, when organosilanes with bulky organic groups (that are anchored in one step) are used, there are probably steric problems that prevent a rapid sealing of the mesopores with the consequent dye loss.

### 3.3. Sensing Studies

Once characterized the material, the response of S1 was tested in the absence and presence of amines by monitoring the release of the entrapped Rhodamine 6G from S1 (λ_ex_ = 525 nm and λ_em_ = 550 nm). In a typical experiment, 1 mg of solid S1 was suspended in 300 µL of a 10^−2^ M aqueous solution of amine (Spm, Spd, and putrescine, respectively), and then this suspension was diluted with 2700 µL of PBS buffer (pH = 7.4) to obtain a 10^−3^ M concentration of amines. As can be seen in Figure 5, a negligible release of the entrapped Rhodamine 6G was observed both in S1 and S1 + putrescine, whereas Spm and Spd induce the gate opening and the release of the dye.

The different results obtained with the studied amines could be related to the pKa values of the compounds involved in the process. In the buffered solution (pH = 7.4), only the less basic primary amino groups of Spm and Spd are not protonated (pK_a_ = 10.9 for both amines) [52]. N-hydroxysuccinimide (pKa = 6.0 [53]) is acid enough to protonate these less-basic amino groups. This acid–base reaction increases shell solubility, giving rise to pore opening. The lower basicity of the primary amines of putrescine could explain the different behavior induced by this compound. Additionally, the higher hydrophilic character of Spm and Spd (lg P = −1.26, −1 for Spm, Spd, respectively) when compared with putrescine (and −0.72) could allow them to dissolve into the organic shell and induce the dye release.

To demonstrate this mechanism, NMR spectra of D_2_O suspensions of S1 in the presence and absence of Spd were registered. Whereas in the absence of the amine, only signals from some residual solvents were observed, in the presence of spermidine, peaks that could be assigned to the molecular gate appear in the solution (see Appendix A in Appendix A).

From titration studies of S1 in the presence of Spm and Spd (Figure 6 for Spm; see Appendix A in Appendix A for Spd), LODs of 27 µM and of 45 µM were calculated for Spm and Spd, respectively.

To evaluate the selectivity of S1, other biological amines were tested (dopamine, GABA, adrenaline, noradrenaline, serotonin, histamine, N-acetylcysteine, tryptophan, alanine, and glycine) and they did not produce any fluorescence (data no show). Additionally, ammonium hydroxide, trimethylamine, 1–5-pentanediamine, propylamine, and pentylamine were also studied, and the results are summarized in Appendix A. In a typical experiment, 2 mg of S1 was added to a 10^−3^ M solution of the corresponding amine in PBS (3 mL). After stirring for 1 h, the mixture was centrifuged, and the fluorescence of the supernatant was observed. In all cases, no increase in fluorescence was observed.

### 3.4. Cell Experiments

To carry out the corresponding studies in cells, RAW 264.7 macrophages were used. Cell viability was determined using the MTT test after 2 h. As can be seen in Appendix A in the Appendix A, S1 does not have any effect on cell viability. On the other hand, the same results were observed with Spm and Spd neither in the absence nor in the presence of S1.

The sensing ability of S1 to detect Spd in RAW 264.7 macrophages was tested. For this purpose, RAW 264.7 macrophages were seeded in 6-well plates (2.0 × 10^6^ cell/well) and maintained overnight in DMEM/F12 medium containing 10% fetal calf serum and 1% penicillin/streptomycin. The cells were incubated with Spd (100 and 200 μg/mL) and kept for 1 h. Then, they were washed with fresh medium and treated with S1 (100 μg/mL) for 1 h. To compare, the same experiment was carried out in the absence of the amine. Figure 7 shows the fluorescence images of the RAW 264.7 cells incubated with S1 and S1+ Spd. An enhancement of the fluorescence was observed, which indicates that S1 is able to detect the amine in cells due to molecular gate opening and entrapped dye release.

## 4. Conclusions

A new hybrid organic–inorganic material (S1) has been prepared and characterized. The material is based on MCM-41 particles functionalized with an N-hydroxysuccinimide derivative. The N-hydroxysuccinimide moiety presents molecular gate properties, and is capable to keep the dye Rhodamine 6G inside the porous material. Two different procedures have been used in the synthetic process. In the first one, the molecular gate is prepared in solution and then bound to the particles, whereas in the second one, the molecular gate is prepared in two consecutive steps. Both materials present similar characteristics. In the presence of Spm or Spd, the molecular gate is opened, and the Rhodamine 6G is released sensing the presence of the amines. Putrescine and other biologically important amines do not act as interferents. The limits of detection calculated using fluorescence are 27 µM and 45 µM for Spm and Spd, respectively. These values indicate that S1 could be used to detect pathologic levels of Spm and Spd in biological fluids. A preliminary study in urine has been developed with positive results, and more exhaustive experiments are in progress to determine the probe utility in other biological fluids. In addition, detection has been carried out in solution and RAW 264.7 macrophages also with interesting results, and tumor cell lines will be studied in the future.

## Data Availability

Not applicable.

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
