# Peer review of "Chromogenic Chemodosimeter Based on Capped Silica Particles to Detect Spermine and Spermidine"

_nanomaterials, 2021, doi:10.3390/nano11030818_

Round 1
Reviewer 1 Report
The authors have very well addressed the reviewers' comments and added substantial supporting data.
I would support its publication.
Author Response
Thank you for your recomendation
Reviewer 2 Report
The resubmitted manuscript was corrected according to some of the provided comments during review of the previous submission. However, a number of comments was not considered. Please introduce the necessary corrections.
- There are still two versions “ml” and “mL”, please unify.
- It would be useful to characterize briefly the observable phases on the plots presented in Figures 4a and 4b as well as indicate the meaning of color curves directly on the plots.
- The grammar of the manuscript was not corrected. Minor typos and grammar errors need to be corrected before publication.
Author Response
We would like to thank the referee for his/her kind comments. Following these suggestions, the manuscript has been carefully revised to unify the style and improve grammar and spelling.
The referee also says “It would be useful to characterize briefly the observable phases on the plots presented in Figures 4a and 4b as well as indicate the meaning of color curves directly on the plots”
We have highlighted in the figure itself the meaning of the black and red lines. In addition, this aspect is also indicated in the figure caption. Additionally, we have included in the figure caption the meaning of the two schematic representations that appear in the figure. Thus, we think that in the current form, there are no doubts in relation to the information contained in the figure.
Reviewer 3 Report
This manuscript can now be accepted for publication.
Author Response
Thank you for your recomendation
Reviewer 4 Report
This paper by Barros and co-workers demonstrates how capped silica particles can be used for the detection of spermine and spermidine. The results are generally interesting but there is no convincing proof that the Nanomaterials journal is an appropriate medium to disseminate these findings if the materials are larger than 100 nm. Please see the suggestions on how this article can be improved.
1) There are some English errors which must be eliminated. Example "What is more, at the best of our knowledge there are not examples of Spm or Spm detection using silica nanoparticles capped with molecular gates". It should read "What is more, at the best of our knowledge there are no examples of Spm or Spm detection using silica nanoparticles capped with molecular gates". Please carefully proofread the work to eliminate such shortcomings.
2) The formatting of the article is inconsistent. Plots are not of the same size and the plots themself have different thicknesses of the curves or they are not alike (some of them have four borders, the other have two). Please make the article consistent to improve its readability.
3) What is the concentration of spermine and spermidine in blood? Readers need to know if these sensors can be used for their direct quantification.
4) These structures are too large to be called nanomaterials "The sample morphology can be described as formed by spherical and pseudo spherical mesoporous nanoparticles with individual sizes lower than 200 nm: with average sizes of 135±45 and 138±53 nm for MCM-41 and S1 samples, respectively". They would need to be smaller than 100 nm to be called like this. Please redefine the name of the material accordingly.
In light of the foregoing, major revision is requested.
Author Response
We would like to thank the referee for his/her kind comment. Following his/her suggestions:
- The manuscript has been carefully revised to unify the style and improve grammar and spelling.
- The formatting of the manuscript has been improved. In that sense, we have try to present all the figures with a similar structure.
- In the future, our interest is to detect these amines in urine and for this reason the value of concentration in urine of ovarian tumors patients has be included in the text.
- We appreciate the careful reading of the reviewer 4. It is true that we did not completely change the term nanoparticle by particle throughout the paper. In the current version, whenever we refer to particles (MCM-41 or S1), we use the term particle (and not nanoparticle). Although the presence of mesopores of ca. 2 nm and ordinates also at the nanoscale, we believe that it would allow us to label the solid S1 as a nanomaterial, according to the criteria of the reviewer, we have changed nanomaterial for material.
Round 2
Reviewer 4 Report
Thank you very much. I am pleased to recommend the paper for publication.